# Similarity Measures of Linguistic Cubic Hesitant Variables for Multiple Attribute Group Decision-Making

**Xueping Lu * and Jun Ye** 

Department of Electrical and Information Engineering, Shaoxing University, 508 Huancheng West Road, Shaoxing 312000, China; yejun@usx.edu.cn
* Correspondence: luxueping@usx.edu.cn

**Abstract:** A linguistic cubic hesitant variable (LCHV) is a hybrid form of linguistic values in group decision-making environments. It is composed of an interval language variable and multiple single-valued language variables given by different decision-makers (DMs). Due to the uncertainty and hesitation of DMs, the numbers of language variables in different LCHVs are unequal. Thus, the least common multiple number (LCMN) extension method was adopted. Based on the included angle and distance of two LCHVs, we presented two cosine similarity measures and developed a multiple attribute group decision-making (MAGDM) approach. An example of engineer selection was used to implement the proposed LCHV MAGDM method and demonstrate the simplicity and feasibility of the proposed method. The sensitivity analysis of weight changes for the two measures showed that the similarity measure based on distance was more stable than the similarity measure based on included angle in this application.

**Keywords:** cosine measure; the least common multiple number (LCMN); linguistic cubic hesitant variable (LCHV); similarity measure; multiple attribute group decision-making

## 1. Introduction

In the age of big data, we use a large amount of data information to solve decision-making (DM) problems in various fields, such as manufacturing domain [1], selection of power generation technology [2], the selection of a transport service provider [3], etc. But not all evaluation information is directly represented by a real number. How do we deal with existing fuzzy information? Thus, decision-making theory and methods are still a critical research topic. Humans are more used to evaluating in linguistic expression than numerical values. Since 1975, the concept of language variables (LVs) was first proposed by Zadeh [4]. Then Herrera [5,6] used linguistic information to solve DM problems [5,6]. Next, scholars introduced various aggregation operations [7–11] to handle linguistic decision information. In some uncertain environments it is difficult for decision-makers to make an assessment in a single-valued LV; they prefer to give the evaluation in an interval language variable rather than a single-value language variable. An interval language variable is called an uncertain linguistic value (ULV) [12]. In order to solve uncertain linguistic DM problems, many uncertain linguistic aggregation operators were induced such as the ULV ordered weighted averaging (ULOWA) operator [13], the UL hybrid geometric mean (ULHGM) operator [14], and the uncertain pure linguistic hybrid harmonic averaging (UPLHHA) operator [15]. After that, the linguistic cubic variable (LCV) was proposed by Ye [16]. The LCV is a hybrid linguistic evaluation form which consists of an interval LV (ULV) and a single-valued LV. An LCV represents a comprehensive evaluation of an attribute given by a group of people. Some LCV aggregation operators were developed such as the LCV weighted

geometric averaging (LCVWGA) operator [16], the LCV weighted arithmetic averaging (LCVWAA) operator [16], the LCV Dombi weighted geometric average (LCVDWGA) operator [17], and LCV Dombi weighted arithmetic average (LCVDWAA) operator [17]. Based on these operators, scholars developed corresponding approaches to solve MAGDM problems under a LCV environment.

Although LCV can express group decision-making information, the LCV cannot express the evaluation values of the group objectively and accurately when the evaluation values given by decision-makers differ greatly. For this reason, Ye [18] put forward another hybrid form of linguistic values and defined it as a linguistic cubic hesitant variable (LCHV). In the case of group decision-making, each of the decision-makers provides an evaluation over an attribute with an interval LV or single-valued LV. We merged all the interval LVs into one interval LV and combined all the different single-valued LVs into a hesitant linguistic set. Then, the interval LV and the hesitant linguistic set (HLS) together constitute a new hybrid linguistic value form, which is called LCHV. Hesitant linguistic sets of different LCHVs may have different lengths. Thus, it is difficult to directly aggregate two LCHVs with different lengths. There are many methods to get uniform lengths for hesitant linguistic sets with different lengths [19–22]. Among them, the least common multiple number (LCMN) of hesitant linguistic elements as the extension size is more objective. According to the LCMN extension method, Ye [18] presented the LCVWGA and LCVWAA aggregation operators, but the methods were implemented in a given weight. The stability of this method has not been tested.

The similarity algorithm is an effective tool to measure the degree of similarity between two fuzzy variables or fuzzy sets. The similarity measure has been widely applied to handle various decision-making information such as trapezoidal fuzzy neutrosophic numbers information [23], simplified neutrosophic hesitant fuzzy set information [24], linguistic neutrosophic numbers information [25], neutrosophic cubic sets information [26], and hesitant linguistic neutrosophic numbers information [27]. So far, there are no other studies on LCHVs except for the WGA and WAA aggregation operators proposed by Ye [18]. In this study, two similarity measures were developed to measure the degree of similarity between two LCHVs and a novel method is presented for solving MAGDM problems with LCHV information.

The rest of this article is organized as follows. In Section 2, we briefly review LCHVs. Based on the distance and included angle of two LCHVs, two cosine similarity measures are proposed in Section 3. Then, based on the cosine similarity measures of the LCHVs, the weighted MAGDM methods are presented in Section 4. An application example is presented to implement the proposed DM method and the influence of hesitation extension on similarity is analyzed in Section 5. A sensitivity analysis to change weights is performed in Section 6. Finally, we conclude the study and put forward future prospects for research.

## 2. The Concept of Linguistic Cubic Hesitant Variables

The LCHV is a new hybrid linguistic variable form. Ye [18] first presented the definition of LCHV as follow.

**Definition 1.** *[18] Let $S = \{L\gamma | \gamma \in [0, \tau]\}$ be an LTS, where $\tau$ is an even number. An LCHV $V$ is defined as $V = (\widetilde{Lu}, \widetilde{Lh})$, where $\widetilde{Lu} = [Lp, Lq]$ for $Lp, Lq \in S$ and $q \geq p$ is an uncertain (interval) LV and $\widetilde{Lh} = \{L\phi\beta | L\phi\beta \in S, \beta = 1, \ldots, k\}$ is a hesitant linguistic variable (HLV). The HLV is a set of k single-valued LVs which are ranked in an ascending order.*

For example, five experts are invited to assess the service management level of a hospital. They give their assessments from a linguistic term set S = {$L_0$ (none), $L_1$ (very low), $L_2$ (low), $L_3$ (slightly low), $L_4$ (moderate), $L_5$ (slightly high), $L_6$ (high), $L_7$ (very high), $L_8$ (super high)}.Two experts give interval LVs [$L_3$, $L_5$] and [$L_4$, $L_6$]. The other three experts give single-valued LVs—$L_5$, $L_3$, and $L_5$. Then we merge two interval LVs into one, such as [$L_3$, $L_6$]. Different single-valued variables form a hesitation set as {$L_3$, $L_5$}. The newly constructed uncertain LV and hesitation set form a linguistic

cubic hesitant variable (LCHV) described as $([L_3, L_6], \{L_3, L_5\})$. The LCHV $([L_3, L_6], \{L_3, L_5\})$ represents the assessments of the five experts.

**Definition 2.** *[18] Let* $V = \left(\widetilde{Lu}, \widetilde{Lh}\right) = \left([Lp, Lq], \{L\phi_1, L\phi_2, \ldots, L\phi_\beta\}\right)$ *for* $Lp, Lq, L\phi_i \in \{L\gamma | \gamma \in [0, \tau]\} (i = 1, 2, \ldots, \beta)$ *be a LCHV. We classify it as follows:*

*(1) if* $\phi_i \in [p, q] (i = 1, 2, \ldots, \beta)$ *for* $p, q \in [0, \tau]$, *then* $V = \left([Lp, Lq], \{L\phi_1, L\phi_2, \ldots, L\phi_\beta\}\right)$ *is an internal LCHV.*

*(2) if* $\phi_i \notin [p, q] (i = 1, 2, \ldots, \beta)$ *for* $p, q, \phi_i \in [0, \tau]$, *then* $V = \left([Lp, Lq], \{L\phi_1, L\phi_2, \ldots, L\phi_\beta\}\right)$ *is an external LCHV.*

## 3. Cosine Measures of LCHVs

In the actual situation, the experts evaluated different objects with different hesitant degrees, so the LCHVs obtained from the experts have various numbers of HLVs. In order to realize the similarity measures between them, it is necessary to extend the HLVs to reach the same number of LVs. We used the LCMN extension method [18] to extend the number of HLVs.

Assume that $Vi = \left(\widetilde{Lui}, \widetilde{Lhi}\right) = \left([Lpi, Lqi], \{L\phi_{i1}, L\phi_{i2}, \ldots, L\phi_{i\alpha_i}\}\right) (i = 1, 2, \ldots, k)$ is a set of LCHVs; $\lambda$ is the LCMN of $(\alpha1, \alpha2, \ldots, \alpha k)$ for $\widetilde{Lhi}$ $(i = 1, 2, \ldots, k)$. Then we can extend the LVs of each LCHVs to the same number as the following extension forms:

$$V_1^e = \left([Lp1, Lq1], \left\{\underbrace{\overbrace{L\phi_{11}^1, L\phi_{11}^2, \ldots, L\phi_{11}^{\lambda/\alpha_1}}^{\lambda/\alpha1}, \underbrace{L\phi_{12}^1, L\phi_{12}^2, \ldots, L\phi_{12}^{\lambda/\alpha_1}}_{\lambda/\alpha_1}, \ldots, \underbrace{L\phi_{1\alpha_1}^1, L\phi_{1\alpha_1}^2, \ldots, L\phi_{1\alpha_1}^{\lambda/\alpha_1}}_{\lambda/\alpha_1}}^{\lambda}\right\}\right),$$

$$V_2^e = \left([Lp2, Lq2], \left\{\underbrace{\overbrace{L\phi_{21}^1, L\phi_{21}^2, \ldots, L\phi_{21}^{\lambda/\alpha_2}}^{\lambda/\alpha_2}, \underbrace{L\phi_{22}^1, L\phi_{22}^2, \ldots, L\phi_{22}^{\lambda/\alpha_1}}_{\lambda/\alpha_2}, \ldots, \underbrace{L\phi_{2\alpha_2}^1, L\phi_{2\alpha_2}^2, \ldots, L\phi_{2\alpha_2}^{\lambda/\alpha_2}}_{\lambda/\alpha_2}}^{\lambda}\right\}\right), \quad (1)$$

$$\ldots,$$

$$V_k^e = \left([Lpk, Lqk], \left\{\underbrace{\overbrace{L\phi_{k1}^1, L\phi_{k1}^2, \ldots, L\phi_{k1}^{\lambda/\alpha_k}}^{\lambda/\alpha k}, \underbrace{L\phi_{k2}^1, L\phi_{k2}^2, \ldots, L\phi_{k2}^{\lambda/\alpha_k}}_{\lambda/\alpha k}, \ldots, \underbrace{L\phi_{k\alpha_k}^1, L\phi_{k\alpha_k}^2, \ldots, L\phi_{k\alpha_k}^{\lambda/\alpha_k}}_{\lambda/\alpha k}}^{\lambda}\right\}\right).$$

**Example 1.** *Suppose* $V_1 = ([L_4, L_5], \{L_4, L_6, L_7\})$ *and* $V_2 = ([L_3, L_5], \{L_4, L_6\})$ *are two LCHVs in the LTS* $S = \{L\gamma | \gamma \in [0, 8]\}$. *Then we can obtain the LCMN* $\lambda = 6$ *according to* $\alpha_1 = 3$ *and* $\alpha_2 = 2$. *Based on Equation (1), the extended forms of the two LCHVs are as follows:*

$V_1^e = ([L_4, L_5], \{L_4, L_4, L_6, L_6, L_7, L_7\})$ *and* $V_2^e = ([L_3, L_5], \{L_4, L_4, L_4, L_6, L_6, L_6\})$.

**Definition 3.** *Let* $V_1 = \left([Lp1, Lq1], \{L\phi_{11}, L\phi_{12}, \ldots, L\phi_{1\lambda}\}\right)$ *and* $V_2 = \left([Lp2, Lq2], \{L\phi_{21}, L\phi_{22}, \ldots, L\phi_{2\lambda}\}\right)$ *be two LCHVs in the LTS* $S = \{L\gamma | \gamma \in [0, \tau]\}$. *The linguistic scale function is* $f(L\gamma) = \frac{\gamma}{\tau}$ *for* $\gamma \in [0, \tau]$. *Then based on the distance and included angle of two LCHVs. The two cosine similarity measures between LCHVs are presented below.*

(1) Cosine similarity measures on the basis of distance

$$
S^D{}_{LCHV}(V_1, V_2) = \frac{1}{2} \left\{ \begin{array}{c} \cos\left(\dfrac{|f(Lp_1)-f(Lp_2)|+|f(Lq_1)-f(Lq_2)|}{4}\pi\right) \\ + \cos\left(\dfrac{\sum\limits_{i=1}^{\lambda}|f(L\phi_{1i})-f(L\phi_{2i})|}{2\lambda}\pi\right) \end{array} \right\}
$$

$$
= \frac{1}{2}\left\{ \cos\left(\frac{|p_1-p_2|+|q_1-q_2|}{4\tau}\pi\right) + \cos\left(\frac{\sum\limits_{i=1}^{\lambda}|\phi_{1i}-\phi_{2i}|}{2\tau\lambda}\pi\right) \right\}
$$

(2)

(2) Cosine similarity measure on the basis of the included angle of the two LCHVs

$$
S^A{}_{LCHV}(V_1, V_2) = \frac{1}{2}\left\{ \begin{array}{c} \dfrac{f(Lp_1)f(Lp_2)+f(Lq_1)f(Lq_2)}{\sqrt{(f(Lp_1))^2+(f(Lq_1))^2}\sqrt{(f(Lp_2))^2+(f(Lq_2))^2}} + \\ \dfrac{\sum\limits_{i=1}^{\lambda}(f(L\phi_{1i})f(L\phi_{2i}))}{\sqrt{\sum\limits_{i=1}^{\lambda}(f(L\phi_{1i}))^2}\sqrt{\sum\limits_{i=1}^{\lambda}(f(L\phi_{2i}))^2}} \end{array} \right\}
$$

$$
= \frac{1}{2}\left\{ \begin{array}{c} \dfrac{p_1\times p_2+q_1\times q_2}{\sqrt{(p_1)^2+(q_1)^2}\sqrt{(p_2)^2+(q_2)^2}} + \\ \dfrac{\sum\limits_{i=1}^{\lambda}\phi_{1i}\times\phi_{2i}}{\sqrt{\sum\limits_{i=1}^{\lambda}(\phi_{1i})^2}\sqrt{\sum\limits_{i=1}^{\lambda}(\phi_{2i})^2}} \end{array} \right\}
$$

(3)

According to the Equations (2) and (3), the cosine similarity measures $S^D(V_1, V_2)$ and $S^A(V_1, V_2)$ satisfy properties (q1)–(q3) as follows:

(q1) $0 \le S^D{}_{LCHV}(V_1, V_2) \le 1, 0 \le S^A{}_{LCHV}(V_1, V_2) \le 1$.

(q2) $S^D{}_{LCHV}(V_1, V_2) = S^D{}_{LCHV}(V_2, V_1), S^A{}_{LCHV}(V_1, V_2) = S^A{}_{LCHV}(V_2, V_1)$.

(q3) If $V_1 = V_2$, then $S^D{}_{LCHV}(V_1, V_2) = 1, S^A{}_{LCHV}(V_1, V_2) = 1$.

**Proof.** We will prove the properties (q1)–(q3) of $S^D{}_{LCHV}(V_1, V_2)$ first.

(q1) Let $x = \left|f(Lp_1) - f(Lp_2)\right| + \left|f(Lq_1) - f(Lq_2)\right|$ for $f(Lp_1), f(Lq_1), f(Lp_2)$,.

$f(Lq_2) \in [0, 1]$ and $Y = \sum\limits_{i=1}^{\lambda}\left|f(L\phi_{1i}) - f(L\phi_{2i})\right|$ for $f(L\phi_{1i}), f(L\phi_{2i}) \in [0, 1]$ where.

$i = 1, 2, \ldots \lambda$. There exists $0 \le X \le 2$ and $0 \le Y \le \lambda$, then $0 \le \cos(x\pi/4) \le 1$ and $0 \le \cos(Y\pi/2\lambda) \le 1$. Thus, we get $0 \le S^D{}_{LCHV}(V_1, V_2) \le 1$.

(q2) It is obvious.

(q3) If $V_1 = V_2$, there exists $f(Lp_1) = f(Lp_2), f(Lq_1) = f(Lq_2), f(L\phi_{1i}) = f(L\phi_{2i})$ for $i = 1, 2, \ldots \lambda$. Then $S^D{}_{LCHV}(V_1, V_2) = 1$ holds.

The properties (q1)–(q3) of $S_{LCHV}{}^A(V_1, V_2)$ can be proved as follows:

(q1) $S^A{}_{LCHV}(V_1, V_2) \ge 0$ is obvious. $S^A{}_{LCHV}(V_1, V_2) \le 1$ can be proved as follows:

Based on the Cauchy–Schwarz formula for inequality:

$(\alpha_1\beta_1 + \alpha_2\beta_2 + \ldots + \alpha_n\beta_n)^2 \le (\alpha_1^2 + \alpha_2^2 + \ldots \alpha_n^2) \times (\beta_1^2 + \beta_2^2 + \ldots \beta_n^2)$, where $(\alpha_1, \alpha_2, \ldots \alpha_n) \in R^n$ and $(\beta_1, \beta_2, \ldots \beta_n) \in R^n$, then the following equality holds.

$f(Lp_1)f(Lp_2) + f(Lq_1)f(Lq_2) \le \sqrt{(f(Lp_1))^2 + (f(Lq_1))^2}\sqrt{(f(Lp_2))^2 + (f(Lq_2))^2}$ and

$\sum\limits_{i=1}^{\lambda}(f(L\phi_{1i})f(L\phi_{2i})) \le \sqrt{\sum\limits_{i=1}^{\lambda}(f(L\phi_{1i}))^2}\sqrt{\sum\limits_{i=1}^{\lambda}(f(L\phi_{2i}))^2}$.

Hence, we can get the following results:

$\dfrac{f(Lp_1)f(Lp_2)+f(Lq_1)f(Lq_2)}{\sqrt{(f(Lp_1))^2+(f(Lq_1))^2}\sqrt{(f(Lp_2))^2+(f(Lq_2))^2}} \le 1$ and

$$\frac{\sum\limits_{i=1}^{\lambda}(f(L\phi_{1i})f(L\phi_{2i}))}{\sqrt{\sum\limits_{i=1}^{\lambda}(f(L\phi_{1i}))^2}\sqrt{\sum\limits_{i=1}^{\lambda}(f(L\phi_{2i}))^2}} \le 1$$

According to Equation (3), we have $S^A{}_{LCHV}(V_1, V_2) \le 1$. Thus, $0 \le S^A{}_{LCHV}(V_1, V_2) \le 1$ holds

(q2) It is obvious.

(q3) If $V_1 = V_2$, there exists $f(Lp_1) = f(Lp_2)$, $f(Lq_1) - f(Lq_2)$, $f(L\phi_{1i}) - f(L\phi_{2i})$ for $i = 1, 2, \ldots \lambda$. Then, we have the following equations.

$$f(Lp_1)f(Lp_2) + f(Lq_1)f(Lq_2) = \sqrt{(f(Lp_1))^2 + (f(Lq_1))^2}\sqrt{(f(Lp_2))^2 + (f(Lq_2))^2} \quad \text{and}$$

$$\sum_{i=1}^{\lambda}(f(L\phi_{1i})f(L\phi_{2i})) = \sqrt{\sum_{i=1}^{\lambda}(f(L\phi_{1i}))^2}\sqrt{\sum_{i=1}^{\lambda}(f(L\phi_{2i}))^2}$$

Hence, $S^A{}_{LCHV}(V_1, V_2) = 1$ holds. □

**Definition 4.** *Let $R = \{V_{r1}, V_{r2}, \ldots, V_{rk}\}$ and $G = \{V_{g1}, V_{g2}, \ldots, V_{gk}\}$ be two HLCV sets, where $V_{ri}$ and $V_{gi}$ (i = 1, 2, … k) are HLCVS in the LTS $S = \{L\gamma|\gamma \in [0, \tau]\}$. If we take into account the weights of the elements $V_{ri}$ and $V_{gi}$ (i = 1, 2, … k), the similarity between R and G can be defined, respectively, as follows:*

$$S^{\omega D}{}_{LCHVS}(R, G) = \sum_{j=1}^{j=k} \omega_j S^D{}_{LCHV}(V_{rj}, V_{gj}) \tag{4}$$

$$S^{\omega A}{}_{LCHVS}(R, G) = \sum_{j=1}^{j=k} \omega_j S^A{}_{LCHV}(V_{rj}, V_{gj}) \tag{5}$$

*where $\omega_j \in [0, 1]$ and $\sum\limits_{j=1}^{k} \omega_j = 1$ for $i = 1, 2, \ldots, k$.*

*In addition, the above two weighted cosine similarity measures $S^{\omega D}{}_{LCHVS}(R, G)$ and $S^{\omega A}{}_{LCHVS}(R, G)$ also have following properties (q1)–(q3):*

*(q1)* $0 \le S^{\omega D}{}_{LCHVS}(R, G) \le 1$, $0 \le S^{\omega A}{}_{LCHVS}(R, G) \le 1$;

*(q2)* $S^{\omega D}{}_{LCHVS}(R, G) = S^{\omega D}{}_{LCHVS}(G, R)$, $S^{\omega A}{}_{LCHVS}(R, G) = S^{\omega A}{}_{LCHVS}(G, R)$;

*(q3) If R = G, then* $S^{\omega D}{}_{LCHVS}(R, G) = 1$, $S^{\omega A}{}_{LCHVS}(R, G) = 1$.

The above proprieties (q1)~(q3) for $S^{\omega D}{}_{LCHVS}(R, G)$ and $S^{\omega A}{}_{LCHVS}(R, G)$ can be proved easily.

## 4. MAGDM Approach Based on Cosine Similarity Measures of LCHVs

Based on cosine similarity measures of LCHVs, we will propose an MAGDM method to solve DM problems with LCHVs information.

Suppose $V = \{V_1, V_2, \ldots, V_n\}$ is the set of *n* alternatives and $A = \{A_1, A_2, \ldots, A_m\}$ is the set of m attributes in a case of MAGMD. When decision-makers make decisions about the $A_j$ attribute of the $V_i$ alternative, each decision-maker can give an interval LV or a single-value LV from the LTS $S = \{L\gamma|\gamma \in [0, \tau]\}$. The evaluation values about the $A_j$ attribute of the $V_i$ alternative given by all decision-makers constitute an LCHV $V_{ij}$ which is described as $V_{ij} = (\widetilde{Lu_{ij}}, \widetilde{Lh_{ij}}) = ([Lp_{ij}, Lq_{ij}], \{L\phi_{ij(1)}, L\phi_{ij(2)}, \ldots, L\phi_{ij(\alpha_{ij})}\})$ (i = 1, 2, … n; j = 1, 2, … m) for the uncertain LV ([Lpij, Lqij]) satisfies qij ≥ pij and the HLV set $\widetilde{L_{hij}}$ is ranked in an ascending order. Hence, all the assessed LCHVs construct a decision matrix $M = (Vij)_{n\times m}$. Furthermore, $\omega_j$ is the weight of attribute $A_j$, where $\omega_j \in [0, 1]$ and $\sum\limits_{j=1}^{m} \omega_j = 1$. Determining the weight values of the attribute $\omega_j$ is important for the objectivity of decision results. Some algorithms have presented to determine weight coefficients such as the analytic hierarchy process (AHP) method [28], the Decision-Making and Trial Evaluation Laboratory (DEMATEL)method [29], the best–worst (BWM) method [30], and the full consistency

(FUCOM) method [31]. Each of them applies to different areas of decision-making. For instance, the BWM was used for the selection of wagons for the internal transport of a logistics company [32] and the location selection for round about construction [33]; the AHP was applied to evaluate university web pages [34], etc.

According to the proposed cosine similarity measures, we developed an MAGDM method of LCHVs using the following steps:

**Step 1.** Suppose $\alpha_{ij}$ is the number of LVs in $\widetilde{L_{hij}}$ for $V_{ij}$. The LCMN of $(\alpha_{1j}, \alpha_{2j}, \ldots, \alpha_{nj})(j = 1, 2, \ldots, m)$ is $\lambda_j$. Extend the HLVs of $(\widetilde{L}_{h1j}, \widetilde{L}_{h2j}, \ldots, \widetilde{L}_{hnj})(j = 1, 2, \ldots, m)$ to reach the same number $\lambda_j$. Based on the extension method mentioned above, we can extend one of the LCHV in matrix $M = (Vij)_{n \times m}$ as the following form:

$$V_{ij}^e = \left( [Lp_{ij}, Lq_{ij}], \left\{ \overbrace{\underbrace{L\phi_{ij(1)}^1, L\phi_{ij(1)}^2, \ldots, L\phi_{ij(1)}^{\lambda_j/\alpha ij}}_{\lambda_j/\alpha ij}, \underbrace{L\phi_{ij(2)}^1, L\phi_{ij(2)}^2, \ldots, L\phi_{ij(3)}^{\lambda_j/\alpha ij}}_{\lambda_j/\alpha ij}, \ldots, \underbrace{L\phi_{ij(\alpha ij)}^1, L\phi_{ij(\alpha ij)}^2, \ldots, L\phi_{ij(\alpha ij)}^{\lambda_j/\alpha ij}}_{\lambda_j/\alpha ij}}^{\lambda_j} \right\} \right).$$

Then, we get the extension matrix below:

$$M^e = \begin{array}{c} V_1{}^e \\ V_2{}^e \\ \cdot \\ \cdot \\ \cdot \\ V_n{}^e \end{array} \left[ \begin{array}{cccc} V_{11}^e & V_{12}^e & \cdots & V_{1m}^e \\ V_{21}^e & V_{22}^e & \cdots & V_{2m}^e \\ \cdot & \cdot & \cdot & \cdot \\ \cdot & \cdot & \cdot & \cdot \\ \cdot & \cdot & \cdot & \cdot \\ V_{n1}^e & V_{n2}^e & \cdots & V_{nm}^e \end{array} \right]$$

**Step 2.** Establish an ideal LCHV set as $V^* = (V_1^*, V_2^*, \ldots V_m^*)$ for $V_j^* = \left( [L_\tau, L_\tau], \left\{ \overbrace{L_\tau, L_\tau, \ldots L_\tau}^{\lambda_j} \right\} \right)$ for $(j = 1, 2, \ldots, m)$.

**Step 3.** According to Equation (2) or Equation (3), the cosine similarity can be measured between $V_{ij}^e$ and $V_j^*$, then obtain $S^D{}_{LCHV}\left( V_{ij}^e, V_j^* \right)$ or $S^A{}_{LCHV}\left( V_{ij}^e, V_j^* \right)$.

**Step 4.** Based on the weight of each attribute $A_j$ ($j = 1, 2, \ldots, m$), calculate the overall weighted cosine similarity as follows:

$$S^{\omega D}{}_{LCHVS}\left( V_i^e, V^* \right) = \sum_{j=1}^{j=m} \omega_j S^D{}_{LCHV}\left( V_{ij}^e, V_j^* \right) \tag{6}$$

$$S^{\omega A}{}_{LCHVS}\left( V_i^e, V^* \right) = \sum_{j=1}^{j=m} \omega_j S^A{}_{LCHV}\left( V_{ij}^e, V_j^* \right) \tag{7}$$

where $\omega_j \in [0, 1]$ and $\sum\limits_{j=1}^{m} \omega_j = 1$.

**Step 5.** Rank the alternatives by the results of the overall weighted cosine similarity measure. The better alternative is the one with the bigger cosine similarity measure result.

**Step 6.** End.

## 5. Illustrative Example

Firstly, we cite an example to demonstrate the proposed MAGDM method in LCHVs is as feasible as the existing method in actual applications. Then we compare and analyze the characteristics of the proposed cosine similarity measures.

### 5.1. Illustrative Example

A computer company is looking for a software engineer. The four candidates, V1, V2, V3, and V4, are selected by the human resources department from all of the applicants. They will be further evaluated from three aspects by five experts. The requirements of the three aspects are innovation capability, work experience, and self-confidence. $w = (0.45, 0.35, 0.2)$ is the weight vector of the three requirements. Then, the five experts evaluate each candidate $Vi$ ($i = 1, 2, 3, 4$) over the three attributes $A_j$ ($j = 1, 2, 3$) by LCHVs. S = {$L_0$ (none), $L_1$ (very low), $L_2$ (low), $L_3$ (slightly low), $L_4$ (moderate), $L_5$ (slightly high), $L_6$ (high), $L_7$ (very high), $L_8$ (super high)}, and is the linguistic term set. The following evaluation matrix consists of the LCHV information given by five experts.

$$D = (V_{ij})_{4\times3} = \begin{array}{c} V_1 \\ V_2 \\ V_3 \\ V_4 \end{array} \left[ \begin{array}{ccc} ([L_4,L_6],\{L_5,L_6\}) & ([L_4,L_6],\{L_4,L_6,L_7\}) & ([L_4,L_7],\{L_5,L_6\}) \\ ([L_3,L_5],\{L_4,L_5,L_6\}) & ([L_5,L_7],\{L_6,L_7\}) & ([L_4,L_6],\{L_4,L_5\}) \\ ([L_5,L_7],\{L_5,L_6\}) & ([L_6,L_7],\{L_4,L_5,L_6\}) & ([L_5,L_7],\{L_4,L_6,L_7\}) \\ ([L_6,L_7],\{L_5,L_6,L_7\}) & ([L_5,L_7],\{L_5,L_7\}) & ([L_4,L_6],\{L_6,L_7\}) \end{array} \right]$$

Thus, the decision-making steps based on the distance cosine similarity measure are as follows:

**Step 1.** According to the number $\alpha_{ij}$ of LVs in $\widetilde{L_{hij}}$ for $V_{ij}$($i = 1,2,3,4; j = 1,2,3$), we can obtain the LCMN of $(\alpha_{1j}, \alpha_{2j}, \ldots, \alpha_{4j})$($j = 1,2,3$) as $\lambda_j = 6$. Then we can extend each of the LCHV in matrix $M = (Vij)_{4\times3}$ in the following form:

$$D^e = (V_{ij})_{4\times3} = \begin{array}{c} V_1^e \\ V_2^e \\ V_3^e \\ V_4^e \end{array} \left[ \begin{array}{ccc} ([L_4,L_6],\{L_5,L_5,L_5,L_6,L_6,L_6\}) & ([L_4,L_6],\{L_4,L_4,L_6,L_6,L_7,L_7\}) & ([L_4,L_7],\{L_5,L_5,L_5,L_6,L_6,L_6\}) \\ ([L_3,L_5],\{L_4,L_4,L_5,L_5,L_6,L_6\}) & ([L_5,L_7],\{L_6,L_6,L_6,L_7,L_7,L_7\}) & ([L_4,L_6],\{L_4,L_4,L_4,L_5,L_5,L_5\}) \\ ([L_5,L_7],\{L_5,L_5,L_5,L_6,L_6,L_6\}) & ([L_6,L_7],\{L_4,L_4,L_5,L_5,L_6,L_6\}) & ([L_5,L_7],\{L_4,L_4,L_6,L_6,L_7,L_7\}) \\ ([L_6,L_7],\{L_5,L_5,L_6,L_6,L_7,L_7\}) & ([L_5,L_7],\{L_5,L_5,L_5,L_7,L_7,L_7\}) & ([L_4,L_6],\{L_6,L_6,L_6,L_7,L_7,L_7\}) \end{array} \right]$$

**Step 2.** Establish the ideal LCHV set as $V^* = (V_1^*, V_2^*, \ldots V_j^*)$. Due to $\lambda_j = 6$ for (j = 1, 2, 3), thus, $V_j^* = ([L_8,L_8],\{L_8,L_8,L_8,L_8,L_8,L_8\})$ for ($j = 1,2,3$). If $\lambda_j$ is unequal, the size of $V_j^*$ is different.

**Step 3.** Calculate the cosine similarity measures between $V_{ij}^e$ and $V_j^*$ based on Equation (2). Suppose $X_1 = V_{11}^e = ([L_4,L_6],\{L_5,L_5,L_5,L_6,L_6,L_6\})$ and $X_2 = V_1^* = ([L_8,L_8],\{L_8,L_8,L_8,L_8,L_8,L_8\})$, we calculate $S^D_{LCHV}(V_{11}^e, V_1^*)$ as below.

$$S^D_{LCHV}(V^e{}_{11}, V^*{}_1) = S^D_{LCHV}(X1,X2) = \frac{1}{2} \left\{ \begin{array}{l} \cos\left( \frac{|f(Lp_1)-f(Lp_2)|+|f(Lq_1)-f(Lq_2)|}{4}\pi \right) \\ + \cos\left( \frac{\sum\limits_{i=1}^{\lambda}|f(L\phi_{1i})-f(L\phi_{2i})|}{2\lambda}\pi \right) \end{array} \right\}$$

$$= \frac{1}{2}\left\{ \cos\left( \frac{|p_1-p_2|+|q_1-q_2|}{4\tau}\pi \right) + \cos\left( \frac{\sum\limits_{i=1}^{\lambda}|\phi_{1i}-\phi_{2i}|}{2\tau\lambda}\pi \right) \right\}$$

$$= \frac{1}{2}\left\{ \cos\left( \frac{|4-8|+|6-8|}{4\times8}\pi \right) + \cos\left( \frac{3|5-8|+3|6-8|}{2\times8\times6}\pi \right) \right\}$$

$$= \frac{1}{2}\left\{ \cos\left( \frac{5}{16}\pi \right) + \cos\left( \frac{5}{32}\pi \right) \right\}$$

$$= 0.8567$$

Similarly, we can get all the values of $S^D{}_{LCHV}\left(V^{ij}{}_e, V^*_j\right)$ for $(i = 1, 2, 3, 4)$ and $(j = 1, 2, 3)$

$$S^D{}_{LCHV}\left(V_{11}{}^e, V^*_1\right) = 0.8567, \; S^D{}_{LCHV}\left(V_{12}{}^e, V^*_2\right) = 0.8642, \; S^D{}_{LCHV}\left(V_{13}{}^e, V^*_3\right) = 0.8819;$$
$$S^D{}_{LCHV}\left(V_{21}{}^e, V^*_1\right) = 0.7693, \; S^D{}_{LCHV}\left(V_{22}{}^e, V^*_2\right) = 0.9404, \; S^D{}_{LCHV}\left(V_{23}{}^e, V^*_3\right) = 0.8022;$$
$$S^D{}_{LCHV}\left(V_{31}{}^e, V^*_1\right) = 0.9029, \; S^D{}_{LCHV}\left(V_{32}{}^e, V^*_2\right) = 0.9194, \; S^D{}_{LCHV}\left(V_{33}{}^e, V^*_3\right) = 0.9104;$$
$$S^D{}_{LCHV}\left(V_{41}{}^e, V^*_1\right) = 0.9404, \; S^D{}_{LCHV}\left(V_{42}{}^e, V^*_2\right) = 0.9239, \; S^D{}_{LCHV}\left(V_{43}{}^e, V^*_3\right) = 0.8942.$$

**Step 4.** Through Equation (6), we can get the overall weighted distance cosine similarity as follows:

$$S^{\omega D}{}_{LCHVS}\left(V^e_1, V^*\right) = \sum_{j=1}^{j=3} \omega_j S^D{}_{LCHV}\left(V^e_{1j}, V^*_j\right) = 0.45 \times 0.8567 + 0.35 \times 0.8642 + 0.2 \times 0.8819 = 0.8644$$

$$S^{\omega D}{}_{LCHVS}\left(V^e_2, V^*\right) = \sum_{j=1}^{j=3} \omega_j S^D{}_{LCHV}\left(V^e_{2j}, V^*_j\right) = 0.45 \times 0.7693 + 0.35 \times 0.9404 + 0.2 \times 0.8022 = 0.8358$$

$$S^{\omega D}{}_{LCHVS}\left(V^e_3, V^*\right) = \sum_{j=1}^{j=3} \omega_j S^D{}_{LCHV}\left(V^e_{3j}, V^*_j\right) = 0.45 \times 0.9029 + 0.35 \times 0.9194 + 0.2 \times 0.9104 = 0.9102$$

$$S^{\omega D}{}_{LCHVS}\left(V^e_4, V^*\right) = \sum_{j=1}^{j=3} \omega_j S^D{}_{LCHV}\left(V^e_{4j}, V^*_j\right) = 0.45 \times 0.9404 + 0.35 \times 0.9239 + 0.2 \times 0.8942 = 0.9254$$

**Step 5.** Using the similarity results, we can rank the four candidates as $v_4 > v_3 > v_1 > v_2$.

Similarly, according to the included angle cosine similarity measure of the two LCHVs in Equation (3), suppose $X1 = V^e_{11} = ([L_4, L_6], \{L_5, L_5, L_5, L_6, L_6, L_6\})$ and $X2 = V^*_1 = ([L_8, L_8], \{L_8, L_8, L_8, L_8, L_8, L_8\})$, we calculate $S^A{}_{LCHV}\left(V_{11}{}^e, V^*_1\right)$ as below.

$$
s^A{}_{LCHV}\left(V_{11}{}^e, V^*_1\right) = S^A{}_{LCHV}(X1, X2) = \frac{1}{2}\left\{ \begin{array}{c} \dfrac{f(Lp_1)f(Lp_2)+f(Lq_1)f(Lq_2)}{\sqrt{(f(Lp_1))^2+(f(Lq_1))^2}\sqrt{(f(Lp_2))^2+(f(Lq_2))^2}} + \\[2ex] \dfrac{\sum\limits_{i=1}^{\lambda}(f(L\phi_{1i})f(L\phi_{2i}))}{\sqrt{\sum\limits_{i=1}^{\lambda}(f(L\phi_{1i}))^2}\sqrt{\sum\limits_{i=1}^{\lambda}(f(L\phi_{2i}))^2}} \end{array} \right\}
$$

$$
= \frac{1}{2}\left\{ \begin{array}{c} \dfrac{p_1 \times p_2 + q_1 \times q_2}{\sqrt{(p_1)^2+(q_1)^2}\sqrt{(p_2)^2+(q_2)^2}} + \\[2ex] \dfrac{\sum\limits_{i=1}^{\lambda}\phi_{1i}\times\phi_{2i}}{\sqrt{\sum\limits_{i=1}^{\lambda}(\phi_{1i})^2}\sqrt{\sum\limits_{i=1}^{\lambda}(\phi_{2i})^2}} \end{array} \right\}
$$

$$
= \frac{1}{2}\left\{ \frac{4 \times 8 + 6 \times 8}{\sqrt{4^2+6^2}\sqrt{8^2+8^2}} + \frac{5 \times 8 + 5 \times 8 + 5 \times 8 + 6 \times 8 + 6 \times 8 + 6 \times 8}{\sqrt{3 \times 5^2 + 3 \times 6^2}\sqrt{6 \times 8^2}} \right\}
$$

$$
= \frac{1}{2}\left\{ \frac{5}{\sqrt{26}} + \frac{33}{\sqrt{1098}} \right\} = 0.9882
$$

In the same way, we can obtain the cosine similarity values of each attribute $S^A{}_{LCHV}\left(V_{ij}{}^e, V^*_j\right)$ for $(i = 1, 2, 3, 4)$ and $(j = 1, 2, 3)$ as following.

$$S^A{}_{LCHV}\left(V_{11}{}^e, V^*_1\right) = 0.9882, \; S^A{}_{LCHV}\left(V_{12}{}^e, V^*_2\right) = 0.9786, \; S^A{}_{LCHV}\left(V_{31}{}^e, V^*_3\right) = 0.9803;$$
$$S^A{}_{LCHV}\left(V_{21}{}^e, V^*_1\right) = 0.9785, \; S^A{}_{LCHV}\left(V_{22}{}^e, V^*_2\right) = 0.9917, \; S^A{}_{LCHV}\left(V_{32}{}^e, V^*_3\right) = 0.9872;$$
$$S^A{}_{LCHV}\left(V_{31}{}^e, V^*_1\right) = 0.9911, \; S^A{}_{LCHV}\left(V_{23}{}^e, V^*_2\right) = 0.9965, \; S^A{}_{LCHV}\left(V_{33}{}^e, V^*_3\right) = 0.9815;$$
$$S^A{}_{LCHV}\left(V_{31}{}^e, V^*_1\right) = 0.9940, \; S^A{}_{LCHV}\left(V_{24}{}^e, V^*_2\right) = 0.9864, \; S^A{}_{LCHV}\left(V_{34}{}^e, V^*_3\right) = 0.9888.$$

by Equation (7), we get the overall weighted cosine similarity values as follows:

$$S^{\omega A}{}_{LCHVS}\left(V_1^e, V^*\right) = \sum_{j=1}^{j=3} \omega_j S^D{}_{LCHV}\left(V_{1j}^e, V_j^*\right) = 0.45 \times 0.9882 + 0.35 \times 0.9786 + 0.2 \times 0.9803 = 0.9833$$

$$S^{\omega A}{}_{LCHVS}\left(V_2^e, V^*\right) = \sum_{j=1}^{j=3} \omega_j S^D{}_{LCHV}\left(V_{2j}^e, V_j^*\right) = 0.45 \times 0.9785 + 0.35 \times 0.9917 + 0.2 \times 0.9872 = 0.9849$$

$$S^{\omega A}{}_{LCHVS}\left(V_3^e, V^*\right) = \sum_{j=1}^{j=3} \omega_j S^D{}_{LCHV}\left(V_{3j}^e, V_j^*\right) = 0.45 \times 0.9911 + 0.35 \times 0.9965 + 0.2 \times 0.9815 = 0.9911$$

$$S^{\omega A}{}_{LCHVS}\left(V_4^e, V^*\right) = \sum_{j=1}^{j=3} \omega_j S^D{}_{LCHV}\left(V_{4j}^e, V_j^*\right) = 0.45 \times 0.9940 + 0.35 \times 0.9864 + 0.2 \times 0.9888 = 0.9903$$

From the above similarity results, we can see that the four alternatives are ranked as $v_3 > v_4 > v_2 > v_1$. The rank order based on the included angle cosine similarity measure of two LCHVs is different from the results based on the distance cosine similarity measure.

*5.2. Related Comparison*

Ye's [18] proposed the LCHVWAA and LCHVWGA aggregation operators of LCHVs. Based on the existing operators proposed by Ye [18] and the cosine similarity measures proposed in this paper, the MAGDM results for the example 1 are shown in the Table 1.

**Table 1.** LCHV[1] MAGDM[2] results.

| MAGDM | Similarity or Score | Ranking Order | The Best |
|---|---|---|---|
| $S^{\omega D}{}_{LCHVs}\left(V_i^e, V^*\right)$ | 0.8644, 0.8358, 0.9102, 0.9254 | $v_4 > v_3 > v_1 > v_2$ | V4 |
| $S^{\omega A}{}_{LCHVs}\left(V_i^e, V^*\right)$ | 0.9833, 0.9849, 0.9911, 0.9903 | $v_3 > v_4 > v_2 > v_1$ | V3 |
| LCHVWAA[3] [18] | 5.3666, 5.3228, 5.8822, 6.1173 | $v_4 > v_3 > v_1 > v_2$ | V4 |
| LCHVWGA[4] [18] | 5.3172, 5.0878, 5.8428, 6.0388 | $v_4 > v_3 > v_1 > v_2$ | V4 |

[1] LCHV= linguistic cubic hesitant variable; [2] MAGDM = multiple attribute group decision making; [3] LCHVWAA = linguistic cubic hesitant variable weighted arithmetic average; [4] LCHVWGA = linguistic cubic hesitant variable weighted geometric average.

We can see from Table 1 that the results based on the distance cosine similarity measure are consistent with the results provided by Ye [18]. But the results based on the included angle cosine similarity measure are different from them.

*5.3. Extension Analysis*

In order to analyze the reason why we get different DM results based on two cosine similarity measure methods, we give two LCHVs $Z_1 = ([L_5, L_7], \{L_4, L_6\})$ and $Z_2 = ([L_4, L_6], \{L_5, L_7\})$. Suppose Z1 and Z2 are two LCHVs of a practical example. Suppose LCMN is 6, then $Z_1^e = ([L_5, L_7], \{L_4, L_4, L_4, L_6, L_6, L_6\})$ and $Z_2^e = ([L_4, L_6], \{L_5, L_5, L_5, L_7, L_7, L_7\})$ are the extensions of Z1 and Z2. To compare the influence of the extension on the two cosine similarity measures, assume $V = ([L_8, L_8], \{L_8, L_8\})$ and $V^* = ([L_8, L_8], \{L_8, L_8, L_8, L_8, L_8, L_8\})$. We list the results of $S^D{}_{LCHV}(Z_i, V)$, $S^A{}_{LCHV}(Z_i, V)$, $S^D{}_{LCHV}(Z_i^*, V^*)$, $S^A{}_{LCHV}(Z_i^*, V^*)$ for $(i = 1, 2)$ in Table 2.

**Table 2.** Similarity measures before and after extension.

| Distance Similarity | Result | Included Angle Similarity | Result |
|---|---|---|---|
| $S^D{}_{LCHV}(Z_1 V)$ | 0.9999 | $S^A{}_{LCHV}(Z_1 V)$ | 0.9831 |
| $S^D{}_{LCHV}(Z_2 V)$ | 0.9999 | $S^A{}_{LCHV}(Z_2, V)$ | 0.9831 |
| $S^D{}_{LCHV}(Z_1^*, V^*)$ | 0.9839 | $S^A{}_{LCHV}(Z_1^*, V^*)$ | 0.9839 |
| $S^D{}_{LCHV}(Z_2^*, V^*)$ | 0.9851 | $S^A{}_{LCHV}(Z_2^*, V^*)$ | 0.9839 |

From the table, we can see that $S^D{}_{LCHV}(Z_1V)$ is equal to $S^D{}_{LCHV}(Z_2V)$ and $S^A{}_{LCHV}(Z_1V)$ is equal to $S^A{}_{LCHV}(Z_2,V)$ too. The ranges of the hesitation and uncertainty variables are interchanged in the two HLVS. The results of two cosine measures are the same before extension. After extension, the included angle cosine similarity measures between ideal LCHV are still the same, but the distance cosine similarity measures between ideal LCHV are different. Moreover, the cosine similarity measures between ideal LCHV based on the included angle are increased with the extension, but the cosine similarity measures between ideal LCHV based on the distance are decreased. In this case, we can see that the cosine similarity measure based on the distance is more sensitive to the extension of LCHVs.

## 6. Sensitivity Analysis to Change Weights

Scholars have proposed various models to analyze the stability of proposed multi-criteria decision-making (MDM) methods [35,36]. As shown in the above section, different ranking results were obtained with two similarity measure MDM methods. We usually choose the best alternative according to the ranking results. The ranking results mostly depend on the values of weight coefficients. We will perform a sensitivity analysis to assess how changes in the weights would change the ranking of the alternatives. The sensitivity analysis is shown through eight scenarios with different weight coefficients, as shown in Table 3.

The ranking results by scenarios is shown in Tables 4 and 5. Results show that rankings of alternatives will change as the weight coefficient changes, as can be seen in the alternatives ranking in Table 4. Similarity measures based on the distance method prefer the alternative V4, because alternative V4 is ranked first in seven out of the eight scenarios. By comparing with the alternatives ranking in Table 5, the alternatives V3 and V4, respectively, are the two best alternatives based on the included angle cosine similarity measure method.

Sensitivity analysis showed that the two similarity measure methods were sensitive to changes in weight. The similarity measure based on the distance method was relatively stable and mostly favored alternative V4. The similarity measure based on the included angle method was more sensitive to changes in weight and the ranking results changed dramatically, but the worst alternative was V1.

**Table 3.** Scenarios with different attribute weights.

| Scenarios | Attribute Weight | | |
|---|---|---|---|
| | A1 | A2 | A3 |
| S-1: Uniform of Weight | 0.33 | 0.33 | 0.33 |
| S-2: Priority of Attribute A1 | 0.8 | 0.1 | 0.1 |
| S-3: Priority of Attribute A2 | 0.1 | 0.8 | 0.1 |
| S-4: Priority of Attribute A3 | 0.1 | 0.1 | 0.8 |
| S-5: Priority of Attribute A1, A2 | 0.4 | 0.4 | 0.2 |
| S-6: Priority of Attribute A2, A3 | 0.2 | 0.4 | 0.4 |
| S-7: Priority of Attribute A1, A3 | 0.4 | 0.2 | 0.4 |
| S-8: Given weight | 0.45 | 0.3 | 0.25 |

**Table 4.** Alternatives ranking for different weight scenarios (distance similarity).

| Alternative | Alternatives Ranking by Scenario | | | | | | | |
|---|---|---|---|---|---|---|---|---|
| | S-1 | S-2 | S-3 | S-4 | S-5 | S-6 | S-7 | S-8 |
| V1 | 3 | 3 | 4 | 3 | 3 | 3 | 3 | 3 |
| V2 | 4 | 4 | 3 | 4 | 4 | 4 | 4 | 4 |
| V3 | 2 | 2 | 2 | 1 | 2 | 2 | 2 | 2 |
| V4 | 1 | 1 | 1 | 2 | 1 | 1 | 1 | 1 |

**Table 5.** Alternatives ranking for different weight scenarios (included angle similarity).

| Alternative | Alternatives Ranking by Scenario | | | | | | | |
|:---:|:---:|:---:|:---:|:---:|:---:|:---:|:---:|:---:|
| | S-1 | S-2 | S-3 | S-4 | S-5 | S-6 | S-7 | S-8 |
| V1 | 4 | 3 | 4 | 4 | 4 | 4 | 4 | 4 |
| V2 | 3 | 4 | 2 | 2 | 3 | 3 | 3 | 3 |
| V3 | 2 | 2 | 1 | 3 | 1 | 1 | 2 | 1 |
| V4 | 1 | 1 | 3 | 1 | 2 | 2 | 1 | 2 |

## 7. Conclusions

Under an MAGDM environment, the distance and included angle cosine similarity measures were firstly applied to deal with LCHV information in this paper. Next, we established a novel MAGDM approach on the basis of the LCMN extension and the cosine similarity measures of LCHVs. Then, a practical example was presented to implement the proposed DM method. Although the approaches can solve the MAGDM problem, the DM results of the two methods were different in the example case. Later, we compared and discussed the impact of the extension on the two cosine similarity measures. Finally, we analyzed the sensitivity of the two methods to weights. We summarize the main highlights of the proposed method below.

(1) Cosine similarity measures based on distance and included angle were used to solve a MAGDM problem with LCHV information for the first time. By using the linguistic scale function, the calculation process of similarity measures was simple and the number of calculations small.

(2) In order to demonstrate the stability of the proposed methods, the sensitivity analysis to weight change was performed. By comparison, the similarity measure based on the distance method was a better fit for the engineer selection case.

(3) Although the LCMN extension method was more objective, this paper provides a preliminary analysis of the influence of hesitation extension on similarity measures.

In the future, we could do more research on LCHV MAGDM. For instance, we can forward aggregation operators or measure methods which are not affected by the degree of hesitation. More models are used to analyze stability of the proposed LCHV MAGDM methods in order that decision-makers choose appropriate methods based on the stability. We also can apply the proposed methods to various fields.

**Author Contributions:** J.Y. proposed the cosine similarity measure operators. X.L. presented the MAGDM methods and comparative analysis. All authors wrote the paper together.

**Funding:** This research received no external funding.

**Conflicts of Interest:** The authors declare no conflict of interest.

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
