# Peer review of "Similarity Measures of Linguistic Cubic Hesitant Variables for Multiple Attribute Group Decision-Making"

_information, doi:10.3390/info10050168_

Round 1

Reviewer 1 Report

Thank you for inviting me as a reviewer for manuscript titled Similarity Measures of Linguistic Cubic Hesitant Variables for Multiple Attribute Group Decision Making.

General comments

This paper presents interesting presentation of similarity measures of LCHV and their application in MCDM. LCHV has a great potential in decision making process, especially group decision making, so this topic in important for investigation. Presented LCHV MCDM model is tested on real world problem - engineer selection. Results are good and well presented. In my opinion the paper is almost ready for the publication. Before final acceptance of the paper, authors should revise the paper according below specific comments.

Specific comments

The paper would be more exiting if you implement below improvements:

- Introduction section presents application of various MCMD approaches in uncertainty environment. But, there is no coherent story between this literature review. The aim of literature review is showing the literature gap and your proposal how to close that gap. For example, Neutrosophic theory (NT) is very important field for decision making. Also, NT closing gap that existing for presentation of uncertainty information with fuzzy theory.

- In introduction section authors should clearly present gap in considered scientific area. Please clearly summarise what specific advantages brings your approach.

- Please, provide relevant references in last two years, authors should refer to: A novel approach for the selection of power generation technology using an linguistic neutrosophic combinative distance-based assessment (CODAS) method: A case study in Libya. Energies, 11(9), 2489,  pp. 1-25.

A sensitivity analysis in MCDM problems: A statistical approach. Decision Making: Applications in Management and Engineering, 1 (2), https://doi.org/10.31181/dmame1802050m.

A multicriteria model for the selection of the transport service provider: A single valued neutrosophic DEMATEL multicriteria model. Decision Making: Applications in Management and Engineering, 1 (2), 121-130. https://doi.org/10.31181/dmame1802128l.

- Use capital letters in heading of all sections.

- Since this is a new approach, authors should (for better understanding and future wide implementation in research) use more numerical examples after definitions/theorems.

- Section 2 – Should be extended and better explained.

- One of important parts of MCDM models is algorithm for determining criteria weights. In that point of view authors should present that algorithms (and their extensions) for determining criteria weights, like AHP/ANP, DEMATEL, BWM, FUCOM, SWARA etc. Authors should consider below relevant references:  The selection of wagons for the internal transport of a logistics company: A novel approach based on rough BWM and rough SAW methods. Symmetry, 9(11), 264,  pp. 1-25.

A comparative empirical study of Analytic Hierarchy Process and Conjoint analysis: Literature review. Decision Making: Applications in Management and Engineering, 1 (2), 153-163. https://doi.org/10.31181/dmame1802160p.

Integration of interval rough AHP and interval rough MABAC methods for evaluating university web pages. Applied Soft Computing, 67, pp. 141-163.

A New Model for Determining Weight Coefficients of Criteria in MCDM Models: Full Consistency Method (FUCOM). Symmetry, 10(9), 393,  pp. 1-22.

The location selection for roundabout construction using Rough BWM -Rough WASPAS approach based on a new Rough Hamy aggregator. Sustainability, 10(8), 2817,  pp. 1-27.

- Section 5 – authors presented impressive results, but authors should provide detail calculations for proposed example. Try in a better way to show the results. Kindly explain your proposed approach steps with the numerical example steps in details. All calculations should be presented. That will help to the readers to better understanding and future implementation this very innovative approach.

- Sections 5.2 and 5.3 should be organized in a better way. I must stress that results are good, but authors should try to focus on numerical comparisons. The case study and the discussion of the results are interesting and appreciated. Provide detail discussion what your model brings more then existing models in decision making literature. What did you bring more? Add more detailed discussion in this section.

- The authors are urged to give more proof and explanation about the validity or practicability of the proposed method. Please, show sensitivity analysis:

Multi-criteria decision making: An example of sensitivity analysis, Serbian journal of management, 11(1), 1-27.

A sensitivity analysis in MCDM problems: A statistical approach. Decision Making: Applications in Management and Engineering, 1 (2), 51-80. https://doi.org/10.31181/dmame1802050m

- Add Managerial implications section of proposed approach.

- Please summarise the advantages and limitations of the proposed method in practical applications.

I must congratulation to the authors for effort they made and provided results. I will review the final version of the paper with pleasure. Once again, congrats to the Author.

Author Response

Dear reviewer

Thank you for your valuable comments. I have revised my article according to your comments.

- Introduction section presents application of various MCMD approaches in uncertainty environment. But, there is no coherent story between this literature review. The aim of literature review is showing the literature gap and your proposal how to close that gap. For example, Neutrosophic theory (NT) is very important field for decision making. Also, NT closing gap that existing for presentation of uncertainty information with fuzzy theory.

Answer: I've added some explanations.

 (In some uncertain environment it is difficult for decision makers to make an assessment in a single - valued LV. they prefer to give the evaluation in an interval language variable rather than a single value language variable. An interval language variable was called as uncertain linguistic value (ULV) [12])

- In introduction section authors should clearly present gap in considered scientific area. Please clearly summarise what specific advantages brings your approach.

Answer: I summarize what specific advantages brings the approach in last section detailedly.

- Please, provide relevant references in last two years, authors should refer to:

A novel approach for the selection of power generation technology using an linguistic neutrosophic combinative distance-based assessment (CODAS) method: A case study in Libya. Energies, 11(9), 2489,  pp. 1-25.

A sensitivity analysis in MCDM problems: A statistical approach.Decision Making: Applications in Management and Engineering, 1 (2), https://doi.org/10.31181/dmame1802050m.

A multicriteria model for the selection of the transport service provider: A single valued neutrosophic DEMATEL multicriteria model. Decision Making: Applications in Management and Engineering, 1 (2), 121-130. https://doi.org/10.31181/dmame1802128l.

Answer: I have referred the above literature in introduction.

- Use capital letters in heading of all sections.

Answer: I have revised all.

- Since this is a new approach, authors should (for better understanding and future wide implementation in research) use more numerical examples after definitions/theorems.

 Answer: for better understanding, I explain Hybrid process of HCLV detailedly.

- Section 2 – Should be extended and better explained.

Answer: I have extended the section 2

- One of important parts of MCDM models is algorithm for determining criteria weights. In that point of view authors should present that algorithms (and their extensions) for determining criteria weights, like AHP/ANP, DEMATEL, BWM, FUCOM, SWARA etc. Authors should consider below relevant references: 

The selection of wagons for the internal transport of a logistics company: A novel approach based on rough BWM and rough SAW methods. Symmetry, 9(11), 264,  pp. 1-25.

A comparative empirical study of Analytic Hierarchy Process and Conjoint analysis: Literature review. Decision Making: Applications in Management and Engineering, 1 (2), 153-163. https://doi.org/10.31181/dmame1802160p.

Integration of interval rough AHP and interval rough MABAC methods for evaluating university web pages. Applied Soft Computing, 67, pp. 141-163.

A New Model for Determining Weight Coefficients of Criteria in MCDM Models: Full Consistency Method (FUCOM). Symmetry, 10(9), 393,  pp. 1-22.

The location selection for round about construction using Rough BWM -Rough WASPAS approach based on a new Rough Hamy aggregator. Sustainability, 10(8), 2817,  pp. 1-27.

Answer: I have learned and referred to the above relevant references and I will present the algorithms for determining criteria weights In an unknown weight decision environment.

- Section 5 – authors presented impressive results, but authors should provide detail calculations for proposed example. Try in a better way to show the results. Kindly explain your proposed approach steps with the numerical example steps in details. All calculations should be presented. That will help to the readers to better understanding and future implementation this very innovative approach.

Answer: I have added detail calculations for proposed example.

- Sections 5.2 and 5.3 should be organized in a better way. I must stress that results are good, but authors should try to focus on numerical comparisons. The case study and the discussion of the results are interesting and appreciated. Provide detail discussion what your model brings more then existing models in decision making literature. What did you bring more? Add more detailed discussion in this section.

Answer: I have revise some minor errors in this section. At present, we can only make a preliminary analysis of the impact of hesitation

variable extension on similarity with a special case. Sensitivity analysis is the key to the stability of the method.

- The authors are urged to give more proof and explanation about the validity or practicability of the proposed method. Please, show sensitivity analysis:

Multi-criteria decision making: An example of sensitivity analysis, Serbian journal of management, 11(1), 1-27.

A sensitivity analysis in MCDM problems: A statistical approach.Decision Making: Applications in Management and Engineering, 1 ( 2), 51-80. https://doi.org/10.31181/dmame1802050m

Answer: We have added sensitivity analysis section.

- Add Managerial implications section of proposed approach.

Answer: In this paper we performed the sensitivity analysis to change weights was performed and compared the stability .So we presented Managerial implications as below in conclusion.

In order to demonstrate the stability of the proposed methods, the sensitivity analysis to change weights was performed. By comparison,Similarity measure based on distance method is more fit for the Engineer selection case.

Please summarise the advantages and limitations of the proposed method in practical applications.

Answer: I summarized the advantages and limitations of the proposed method in last section

     Thank you once again. I wish you would review this version with pleasure.

                                                            Xueping Lu

Reviewer 2 Report

While the paper already a well-written and well-structured work with interesting outcomes, in the following I list a few specific points that the authors could implement in order to improve the paper:

1. The abstract should follow the style of structured abstracts: 1) Background: Place the question addressed in a broad context and highlight the purpose of the study; 2) Methods: Describe briefly the main methods or treatments applied. 3) Results: Summarize the article's main findings; and 4) Conclusion: Indicate the main conclusions or interpretations. In the current abstract, the authors give a lot of attention to points 1) and 2), whereas points 3) and 4) are omitted.

2. The Conclusion section should be improved. Please highlight and extend main conclusions regarding the proposed similarity measures and the developed MAGDM method.

3. The manuscript should be carefully checked in terms of typos errors (e.g. line 68, 71, etc.), as well as correct formatting (e.g. lines 162-169, 176-178, etc.).

Author Response

Dear reviewer

Thank you for your valuable comments. I have revised my article according to your comments.

1.       The abstract should follow the style of structured abstracts: 1) Background: Place the question addressed in a broad context and highlight the purpose of the study; 2) Methods: Describe briefly the main methods or treatments applied. 3) Results: Summarize the article's main findings; and 4) Conclusion: Indicate the main conclusions or interpretations. In the current abstract, the authors give a lot of attention to points 1) and 2), whereas points 3) and 4) are omitted.

Response: I have improved the abstract and add the point 3&4.

2.       The Conclusion section should be improved. Please highlight and extend main conclusions regarding the proposed similarity measures and the developed MAGDM method.

Response: I have extended the conclusion and present the highlights of the proposed method

3.       The manuscript should be carefully checked in terms of typos errors (e.g. line 68, 71, etc.), as well as correct formatting (e.g. lines 162-169, 176-178, etc.).

Response: I have checked the typos errors carefully.

Thank you once again. I wish you would review this version with pleasure.

                                                            Xueping Lu

Round 2

Reviewer 1 Report

I read through the paper and thought most of my concerns were well addressed, thus it was acceptable from my side.

Author Response

  Dear reviewer

       I will check my final version carefully.tahk you very much!